# IMPLICIT $\lambda$-JEFFREYS AUTOENCODERS: TAKING THE BEST OF BOTH WORLDS

## ABSTRACT

We propose a new form of an autoencoding model which incorporates the best properties of variational autoencoders (VAE) and generative adversarial networks (GAN). It is known that GAN can produce very realistic samples while VAE does not suffer from mode collapsing problem. Our model optimizes $\lambda$-Jeffreys divergence between the model distribution and the true data distribution. We show that it takes the best properties of VAE and GAN objectives. It consists of two parts. One of these parts can be optimized by using the standard adversarial training, and the second one is the very objective of the VAE model. However, the straightforward way of substituting the VAE loss does not work well if we use an explicit likelihood such as Gaussian or Laplace which have limited flexibility in high dimensions and are unnatural for modelling images in the space of pixels. To tackle this problem we propose a novel approach to train the VAE model with an implicit likelihood by an adversarially trained discriminator. In an extensive set of experiments on CIFAR-10 and TinyImagent datasets, we show that our model achieves the state-of-the-art trade-off between generation and reconstruction quality and demonstrate how we can balance between mode-seeking and mass-covering behaviour of our model by adjusting the weight $\lambda$ in our objective.

## 1 INTRODUCTION

Variational autoencoder (VAE) (Kingma et al., 2014; Rezende et al., 2014; Titsias & Lázaro-Gredilla, 2014) is one of the most popular approaches for modeling complex high-dimensional distributions. It has been applied successfully to many practical problems. It has several nice properties such as learning low-dimensional representations for the objects and ability to conditional generation. Due to an explicit reconstruction term in its objective, one may ensure that VAE can generate all objects from the training set. These advantages, however, come at a price. It is a known fact that VAE tends to generate unrealistic objects, e.g., blurred images. Such behaviour can be explained by the properties of a maximum likelihood estimation (MLE) which is used to fit a restricted VAE model $p_\theta(x)$ in data that comes from a complex distribution $p^*(x)$. We can equivalently reformulate this MLE objective as a minimization of the forward KL divergence $D_{\mathrm{KL}}(p^*(x)\|p_\theta(x))$ which does not explicitly penalize the model $p_\theta(x)$ for generating unrealistic objects (Arjovsky & Bottou, 2017). As a result, many regions with the low value $p^*(x)$ may have a high value of $p_\theta(x)$ in the case when the model $p_\theta(x)$ has the limited capacity (see Figure 1a)).

Another popular generative model is a generative adversarial network (GAN) (Goodfellow et al., 2014) which is known for its ability to sample realistic objects. However, it suffers from an inability to cover the whole distribution $p^*(x)$ that leads to mode-seeking behavior, also known as mode-collapse (Salimans et al., 2016; Metz et al., 2016; Goodfellow, 2016). The main reason of such behaviour is the properties of the reverse KL divergence $D_{\mathrm{KL}}(p_\theta(x)\|p^*(x))$ (or the Jensen-Shanon divergence $\mathrm{JSD}(p_\theta(x)\|p^*(x))$) that is minimized during GAN training. These divergences $D_{\mathrm{KL}}(p_\theta(x)\|p^*(x))$ and $\mathrm{JSD}(p_\theta(x)\|p^*(x))$ do not penalize the model $p_\theta(x)$ for ignoring some high value regions of $p^*(x)$ (Arjovsky & Bottou, 2017). As a result, the most probability mass of the restricted model $p_\theta(x)$ can be concentrated in a small number of modes of $p^*(x)$ (see Figure 1b), 1c)).

In this paper, we propose a new form of an autoencoding model which is based on incorporating of VAE and GAN objectives. We consider $\lambda$-Jeffreys divergence $\mathrm{J}_\lambda(p_\theta(x)\|p^*(x))$ which is a weighted sum of forward and reverse KL divergences: $\mathrm{J}_\lambda(p_\theta(x)\|p^*(x)) = \lambda D_{\mathrm{KL}}(p^*(x)\|p_\theta(x)) + (1 -$

Figure 1: Comparison of $\lambda$-Jeffreys for $\lambda = 0.5$ (red), Reverse KL (green), Forward KL (blue) and JSD (orange) divergences on the task of approximating a mixture of 4 Gaussians (black dashed line) with an equiprobable mixture of two Gaussians with learnable location and scale. Plots a)-c) show pairwise comparisons of optimal log-densities, the plot d) compares optimal densities themselves.

$\lambda)D_{\mathrm{KL}}(p_\theta(x)\|p^*(x))$. This way, we encourage our model to be mode-seeking while still having relatively high values of $p_\theta(x)$ on all objects from a training set, thus preventing the mode-collapse. We note that $\mathrm{J}_\lambda(p_\theta(x)\|p^*(x))$ is not symmetric with respect to $p_\theta(x)$ and $p^*(x)$ and by the weight $\lambda$ we can balance between mode-seeking and mass-covering behaviour.

However, the straightforward way of substituting each KL term with GAN and VAE losses does not work well in practice if we use an explicit likelihood for object reconstruction in VAE objective. Such simple distributions as Gaussian or Laplace that are usually used in VAE have limited flexibility and are unnatural for modelling images in the space of pixels. To tackle this problem we propose a novel approach to train the VAE model in an adversarial manner. We show how we can estimate the implicit likelihood in our loss function by an adversarially trained discriminator.

We theoretically analyze the introduced loss function and show that under assumptions of optimal discriminators, our model minimizes the $\lambda$-Jeffreys divergence $\mathrm{J}_\lambda(p_\theta(x)\|p^*(x))$ and we call our method as Implicit $\lambda$-Jeffreys Autoencoder ($\lambda$-IJAE). In an extensive set of experiments, we evaluate the generation and reconstruction ability of our model on CIFAR10 (Krizhevsky et al., 2009) and TinyImagenet datasets. It shows the state-of-the-art trade-off between generation and reconstruction quality. We demonstrate how we can balance between the ability of generating realistic images and the reconstruction ability by changing the weight $\lambda$ in our objective. Based on our experimental study we derive a default choice for $\lambda$ that establishes a reasonable compromise between mode-seeking and mass-covering behaviour of our model and this choice is consistent over these two datasets.

## 2 RELATED WORK

**Relation to forward KL-based methods.** We can say that all VAE-based models minimize the upper bound on the forward KL $D_{\mathrm{KL}}(p^*(x)\|p_\theta)$. In recent years there have been many extensions and improvements of the standard VAE. One direction of research is to inroduce the discriminator as a part of data likelihood (Larsen et al., 2015; Brock et al., 2017) to leverage its intermediate layers for measuring similarity between objects. However, these models do not have a sound theoretical justification about what distance between $p_\theta(x)$ and $p^*(x)$ they optimize. The other way is to consider more complex variational distribution $q_\varphi(z|x)$. One can either use better variational bounds (Agakov & Barber, 2004; Maaløe et al., 2016; Ranganath et al., 2016; Molchanov et al., 2018; Sobolev & Vetrov, 2019) or apply the adversarial training to match $q_\varphi(z|x)$ and the prior distribution $p(z)$ (Mescheder et al., 2017) or to match the marginals $q_\varphi(z)$ and $p(z)$ (Makhzani et al., 2016). Although these methods improve approximate inference in VAE model, they remain in the scope of MLE framework. As we discussed above within this framework the model with a limited capacity is going to have the mass-covering behaviour.

**Relation to reverse KL-based methods.** The vanilla GAN framework is equivalent under the assumption of optimal discriminator to minimization of Jensen-Shanon divergence $\mathrm{JSD}(p^*(x)\|p_\theta(x))$ (Goodfellow et al., 2014). With a minor modification of a generator loss we can obtain the equivalence to minimization of the reverse KL $D_{\mathrm{KL}}(p_\theta(x)\|p^*(x))$ (Huszar, 2016; Arjovsky & Bottou, 2017). There have been proposed many autoencoder models which utilize one of these two divergences $\mathrm{JSD}(p^*(x)\|p_\theta(x))$ and $D_{\mathrm{KL}}(p_\theta(x)\|p^*(x))$. One approach is to minimize the divergence between joint distributions $p^*(x)q(z|x)$ and $p_\theta(x|z)p(z)$ in a GAN framework (Donahue et al., 2017; Dumoulin et al., 2017). ALICE model (Li et al., 2017) introduces an additional entropy loss for dealing with the non-identifiability issues in previous works. Other methods (Chen et al., 2018;

Pu et al., 2017a; Rosca et al., 2017; Ulyanov et al., 2018; Zhu et al., 2017) use the reverse KL $D_{\mathrm{KL}}(p_\theta(x)\|p^*(x))$ as an additional term to encourage mode-seeking behaviour.

**Relation to Jeffreys divergence-based methods.** To the best of our knowledge, there are only two other autoencoder models which minimize $\lambda$-Jeffreys divergence for $\lambda = 0.5$. It is an important case when $\lambda$-Jeffreys divergence equals symmetric KL divergence. These methods are AS-VAE (Pu et al., 2017a) and SVAE (Chen et al., 2018) and they are most closely related to our work. AS-VAE is a special case of SVAE method therefore further we will consider only SVAE. There are two most crucial differences between SVAE and $\lambda$-IJAE models. The first one is that SVAE minimizes $J_\lambda(p^*(x)q(z|x)\|p_\theta(x|z)p(z))$ between joint distributions $p^*(x)q(z|x)$ and $p_\theta(x|z)p(z)$ for $\lambda = 0.5$ while $\lambda$-IJAE minimizes $J_\lambda(p^*(x)\|p_\theta(x))$ between marginal distributions $p^*(x)$ and $p_\theta(x)$ for arbitrary $\lambda$. The second difference is that the SVAE's loss $J_\lambda(p^*(x)q(z|x)\|p_\theta(x|z)p(z))$ solely did not give good reconstructions in experiments. Therefore, authors introduced additional data-fit terms $\mathbb{E}_{p^*(x)q_\theta(z|x)} \log p_\theta(x|z) + \mathbb{E}_{p_\theta(x|z)p(z)} \log q_\varphi(z|x)$ where $p_\theta(x|z)$ and $q_\varphi(z|x)$ are explicit densities. In contrast, $\lambda$-IJAE model achieves good generation and reconstruction quality as it is and allows training implicit $p_\theta(x|z)$ and $q_\varphi(z|x)$ distributions. These two differences make SVAE and $\lambda$-IJAE models significantly distinct, and we observe it in practice.

## 3 Background

Consider samples $x \in \mathcal{X}$ from the true data distribution $p^*(x)$. The aim of generative models is to fit a model distribution $p_\theta(x)$ to $p^*(x)$. Most popular models are GAN and VAE. In practice, we observe that they have significantly different properties. VAE tends to cover all modes of $p^*(x)$ at the cost of capturing low probability regions as well. As a result, it often generates unspecific and/or blurry images. On the other hand, GAN is highly mode-seeking, i.e. it tends to concentrate most of its probability mass in a small number of modes of $p^*(x)$. Therefore it may not cover significant part of $p^*(x)$ which is also known as a mode collapse problem. Such radical contrast between VAE and GAN can be explained by the fact that they optimize different divergences between $p_\theta(x)$ and $p^*(x)$.

**Variational Inference.** VAE is trained by MLE: $\max_\theta \mathbb{E}_{p^*(x)} \log p_\theta(x)$. The distribution $p_\theta(x)$ is defined as an integral over a latent variable $z$: $p_\theta(x) = \int p_\theta(x|z)p(z)dz$, and in practice it is typically intractable. Variational inference (Hinton & Van Camp, 1993) sidesteps this issue by introducing an encoder model (also known as a variational distribution) $q_\varphi(z|x)$ and replacing the intractable $\log p_\theta(x)$ with a tractable *evidence lower bound* (ELBO):

$$\mathbb{E}_{p^*(x)} \log p_\theta(x) \geqslant \mathbb{E}_{p^*(x)} \mathbb{E}_{q_\varphi(z|x)} \log \frac{p_\theta(x|z)p(z)}{q_\varphi(z|x)} =$$
$$= \mathbb{E}_{p^*(x)} \left[ \mathbb{E}_{q_\varphi(z|x)} \log p_\theta(x|z) - KL(q_\varphi(z|x)\|p(z)) \right] = \mathcal{L}_{\mathrm{ELBO}}(\theta, \varphi) \quad (1)$$

Then we maximize ELBO $\mathcal{L}_{\mathrm{ELBO}}(\theta, \varphi)$ with respect to $\theta$ and $\varphi$. One can easily derive that MLE is equivalent to optimizing the forward KL $D_{\mathrm{KL}}(p^*\|p_\theta)$:

$$\theta^* = \arg\max_\theta \left[ \mathbb{E}_{p^*(x)} \log p_\theta(x) \right] = \arg\max_\theta \left[ -\mathbb{E}_{p^*(x)} \log \frac{p^*(x)}{p_\theta(x)} \right] = \arg\min_\theta D_{\mathrm{KL}}(p^*\|p_\theta) \quad (2)$$

**Adversarial Training.** The adversarial framework is based on a game between a generator $G_\theta(z)$ and a discriminator $D_\psi(x)$ which classifies objects from $p^*(x)$ versus ones from $p_\theta(x)$:

$$\min_\theta \max_\psi \left[ \mathbb{E}_{p^*(x)} \log D_\psi(x) + \mathbb{E}_{p_\theta(x)} \log(1 - D_\psi(x)) \right] \quad (3)$$

Goodfellow et al. (2014) showed that the loss of the generator (3) is equivalent to the Jensen-Shanon divergence $\mathrm{JSD}(p_\theta\|p^*)$ given an optimal discriminator $D_{\psi^*}(x) = \frac{p^*(x)}{p^*(x)+p_\theta(x)}$:

$$\mathbb{E}_{p^*(x)} \log D_{\psi^*}(x) + \mathbb{E}_{p_\theta(x)} \log(1 - D_{\psi^*}(x)) = D_{\mathrm{KL}} \left( p^* \left\| \frac{p^*+p_\theta}{2} \right. \right) + D_{\mathrm{KL}} \left( p_\theta \left\| \frac{p^*+p_\theta}{2} \right. \right) - \log 4$$

It is easy to recognize this as an instance of classification-based Density Ratio Estimation (DRE) (Sugiyama et al., 2012). Following this framework, one can consider different generator's objectives while keeping the same objective (3) for the discriminator. DRE relies on the fact that $\frac{D_{\psi^*}(x)}{1-D_{\psi^*}(x)} =$

$\frac{p^*(x)}{p_\theta(x)}$. By this approach we can obtain a likelihood-free estimator for the reverse $D_{\mathrm{KL}}(p_\theta \| p^*)$ (Huszar, 2016):

$$-\mathbb{E}_{p_\theta(x)} \log \frac{D_{\psi^*}(x)}{1 - D_{\psi^*}(x)} = \mathbb{E}_{p_\theta(x)} \log \frac{p_\theta(x)}{p^*(x)} = D_{\mathrm{KL}}(p_\theta \| p^*)$$

**(Un)Biased Gradients in Adversarial Training.** Since in practice the discriminator $D_\psi(x)$ is only trained to work for one particular set of generator parameters $\theta$, we need to be cautious regarding validity of gradients obtained by DRE approach. For example, consider the forward KL $D_{\mathrm{KL}}(p^* \| p_\theta)$. If we apply DRE, we will arrive at $\mathbb{E}_{p^*(x)} \log \frac{D_\psi(x)}{1 - D_\psi(x)}$. However, we can notice that in practice this expression does not depend on $\theta$ in any way, i.e. $\nabla_\theta \mathbb{E}_{p^*(x)} \log \frac{D_\psi(x)}{1 - D_\psi(x)} = 0$. This is because the forward KL depends on $\theta$ only through the ratio of densities, which is replaced by a point estimate using a discriminator which has no idea regarding $p_\theta$'s local behaviour.

This shows we need to be careful when designing adversarial learning objective as to ensure unbiased gradients. Luckily, $\mathrm{JSD}(p_\theta \| p^*)$ and $D_{\mathrm{KL}}(p_\theta \| p^*)$ are not affected by this problem:

**Proposition 1.** *(Mescheder et al., 2017) Let* $D_{\psi^*}(x) = \frac{p^*(x)}{p^*(x) + p_\theta(x)}$ *for any* $x$. *Then* $-\nabla_\theta \mathbb{E}_{p_\theta(x)} \log \frac{D_{\psi^*}(x)}{1 - D_{\psi^*}(x)} = \nabla_\theta D_{\mathrm{KL}}(p_\theta \| p^*)$ *and* $\nabla_\theta \mathbb{E}_{p_\theta(x)} \log(1 - D_{\psi^*}(x)) = \nabla_\theta \mathrm{JSD}(p_\theta \| p^*)$ *even if we assume that* $\nabla_\theta D_{\psi^*}(x) = 0$.

*Proof.* Given in Appendix, section A. □

## 4 Implicit $\lambda$-Jeffreys Autoencoder

VAE provide a theoretically sound way to learn generative models with a natural and coherent encoder. However, they are known to generate blurry and unspecific samples that have inferior perceptual quality compared to generative models based on adversarial learning. The main cause for that is that the root principle VAEs are built upon – MLE framework – is equivalent to minimization of the forward KL $D_{\mathrm{KL}}(p^* \| p_\theta)$. While $D_{\mathrm{KL}}(p^* \| p_\theta)$ recovers the true data-generating process $p^*(x)$ if the model $p_\theta(x)$ has enough capacity, in a more realistic case of an insufficiently expressive model $p_\theta(x)$ it is known to be mass-covering. As a result, the model is forced to cover all modes of $p^*(x)$ even at the cost of covering low-probability regions as well. This in turn might lead to blurry samples as the model does not have the capacity to concentrate inside the modes. On the other hand, the reverse KL $D_{\mathrm{KL}}(p_\theta \| p^*)$ has mode-seeking behavior that penalizes covering low-probability regions and thus the model $p_\theta(x)$ tends to cover only a few of the modes of $p^*(x)$.

Following this reasoning, we propose a more balanced divergence – one that seeks modes, but still does a decent job covering all modes of $p^*(x)$ to prevent mode collapse. We chose $\lambda$-Jeffreys divergence (Jeffreys, 1998) between $p_\theta(x)$ and $p^*(x)$: $\mathrm{J}_\lambda(p_\theta(x) \| p^*(x)) = \lambda D_{\mathrm{KL}}(p^*(x) \| p_\theta(x)) + (1 - \lambda) D_{\mathrm{KL}}(p_\theta(x) \| p^*(x))$.

We illustrate the advantage of $\lambda$-Jeffreys divergence for $\lambda = 0.5$ over Forward KL, Reverse KL and JSD divergences in the case of a model with limited capacity in Figure 1. In this figure we compared divergences in a simple task (see Appendix, section B) of approximating a mixture of 4 Gaussians with a mixture of just two: both Reverse KL and JSD exhibit mode-seeking behavior, completely dropping side modes, whereas the Forward KL assigns much more probability to tails and does poor job capturing the central modes. On a contrast, $\lambda$-Jeffreys divergence uses one mixture component to capture the most probable mode, and the other to ensure mass-covering.

The optimization of $\lambda$-Jeffreys divergence consists of two parts. The first one is the minimization of the reverse KL $D_{\mathrm{KL}}(p_\theta(x) \| p^*(x))$ which can be implemented as a standard GAN optimization as we discussed in Section 3. The second part is the optimization of the forward KL $D_{\mathrm{KL}}(p^*(x) \| p_\theta(x))$ and we tackle it by maximization of the ELBO $\mathcal{L}_{\mathrm{ELBO}}(\theta, \varphi)$ as in VAE. So, we obtain an upper bound on $\lambda$-Jeffreys divergence by incorporating GAN and VAE objectives:

$$\mathcal{L}_{\lambda\text{-IJAE}}(\theta, \varphi) = (1 - \lambda) D_{\mathrm{KL}}(p_\theta(x) \| p^*(x)) - \lambda \mathcal{L}_{\mathrm{ELBO}}(\theta, \varphi) \geqslant \mathrm{J}_\lambda(p_\theta(x) \| p^*(x))$$

The ELBO term $\mathcal{L}_{\mathrm{ELBO}}(\theta, \varphi)$ can be decomposed into two parts: (*i*) a reconstruction term $\mathbb{E}_{p^*(x)} \mathbb{E}_{q_\varphi(z|x)} \log p_\theta(x|z)$; (*ii*) a KL term $\mathbb{E}_{p^*(x)} KL(q_\varphi(z|x) \| p(z))$. While both terms are easy

to deal with in cases of explicit $p(x|z)$ and $q(z|x)$, an implicit formulation poses some challenges. In the next two sections we address them.

**Implicit Conditional Likelihood.** Typically to optimize the reconstruction term $\mathbb{E}_{p^*(x)}\mathbb{E}_{q_\varphi(z|x)} \log p_\theta(x|z)$ the conditional likelihood $p_\theta(x|z)$ is defined explicitly as a fully factorized Gaussian or Laplace distribution (Kingma et al., 2014; Rezende et al., 2014; Titsias & Lázaro-Gredilla, 2014; Pu et al., 2017b; Chen et al., 2018; Rosca et al., 2017; Mescheder et al., 2017). While convenient, such choice might limit the expressivity of the generator $G_\theta(z)$. As we discussed previously, optimization of the forward $KL(p^*\|p_\theta)$ leads to a mass-covering behavior. The undesired properties of this behavior such as sampling unrealistic and/or blurry images can be more significant if a capacity of our model $p_\theta(x)$ is limited. Therefore we propose a technique which allows to extend the class of possible likelihoods for $p_\theta(x|z)$ to implicit ones.

We note that typically in VAE the decoder $p_\theta(x|z)$ first maps the latent code $z$ to the space $\mathcal{X}$, which is then used to parametrize the distribution of $z$'s decodings $x|z$. For example, this is the case for $\mathcal{N}(x|G_\theta(z), \sigma I)$ or $Laplace(x|G_\theta(z), \sigma I)$. We also use the output of the generator $G_\theta(z) \in \mathcal{X}$ to parametrize an implicit likelihood. In particular, we assume $p_\theta(x|z) = r(x|G_\theta(z))$ for some *symmetric likelihood* $r(x|y)$:

**Definition 1.** *A density $r(\cdot|\cdot) : \mathcal{X} \times \mathcal{X} \to \mathbb{R}_+$ is a symmetric likelihood if*

  *(i) $r(x = a|y = b) = r(x = b|y = a) \quad \forall a, b \in \mathcal{X}$;*

  *(ii) $r(x = a|y = b)$ has a mode at $a = b$.*

While the Gaussian and Laplace likelihoods are symmetric and explicit, in general we do not require $r(x|y)$ to be explicit, only being able to generate samples from $r(x|y)$ is required.

The idea is to introduce a discriminator $D_\tau(x, z, y)$ which classifies two types of triplets:

  - real class: $(x, z, y) \sim p^*(x)q_\varphi(z|x)r(y|x)$;
  - fake class: $(x, z, y) \sim p^*(x)q_\varphi(z|x)r'(y|G_\theta(z))$.

We note that $r(y|x)$ and $r'(y|x)$ can be different and we will utilize this possibility in practice. Then we train the discriminator $D_\tau(x, z, y)$ using the standard binary cross-entropy objective:

$$\mathbb{E}_{p^*(x)q_\varphi(z|x)} \left[ \mathbb{E}_{r(y|x)} \log D_\tau(x, z, y) + \mathbb{E}_{r'(y|G_\theta(z))} \log(1 - D_\tau(x, z, y)) \right] \to \max_\tau \quad (4)$$

If we apply the straightforward way to obtain an objective for the generator $G_\theta(z)$ we will derive that we should minimize $\mathbb{E}_{p^*(x)q_\varphi(z|x)} \log \frac{D_\tau(x,z,x)}{1-D_\tau(x,z,x)}$. Indeed, given an optimal discriminator for (4) $D_{\tau^*}(x, z, y) = \frac{r(y|x)}{r(y|x)+r'(y|G_\theta(z))}$, we obtain:

$$\mathbb{E}_{p^*(x)q_\varphi(z|x)} \log \frac{D_{\tau^*}(x, z, x)}{1 - D_{\tau^*}(x, z, x)} = \mathbb{E}_{p^*(x)q_\varphi(z|x)} \log \frac{r(x|x)}{r'(x|G_\theta(z))} =$$
$$= -\mathbb{E}_{p^*(x)q_\varphi(z|x)} \log r'(x|G_\theta(z)) + \text{Const}$$

So, we see that minimizing $\mathbb{E}_{p^*(x)q_\varphi(z|x)} \log \frac{D_\tau(x,z,x)}{1-D_\tau(x,z,x)}$ given the optimal $D_{\tau^*}(x, z, y)$ is equivalent to maximizing the reconstruction term with $p_\theta(x|z) = r'(x|G_\theta(z))$. However, we face in practice the same issue as we discussed in Section 3 that $\nabla_\theta \mathbb{E}_{p^*(x)q_\varphi(z|x)} \log \frac{D_\tau(x,z,x)}{1-D_\tau(x,z,x)} = 0$ because $D_\tau(x, z, x)$ does not depend on $\theta$ explicitly even for optimal $\tau = \tau^*$.

We can overcome this issue by exploiting the properties of symmetric likelihoods if we minimize a slightly different loss for the generator $G_\theta(z)$: $-\mathbb{E}_{p^*(x)q_\varphi(z|x)} \log \frac{D_\tau(x,z,G_\theta(z))}{1-D_\tau(x,z,G_\theta(z))}$. The following theorem guarantees the gradients will be unbiased in the optimal discriminator case:

**Theorem 1.** *Let $D_{\tau^*}(x, z, y)$ be the optimal solution for the objective (4) and $r(y|x)$ and $r'(y|x)$ are symmetric likelihoods. Then $\nabla_\theta \mathbb{E}_{p^*(x)} \mathbb{E}_{q_\varphi(z|x)} \log \frac{D_{\tau^*}(x,z,G_\theta(z))}{1-D_{\tau^*}(x,z,G_\theta(z))} = \nabla_\theta \mathbb{E}_{p^*(x)} \mathbb{E}_{q_\varphi(z|x)} \log r(x|G_\theta(z))$.*

*Proof.* Given in Appendix, section A. $\square$

So, we obtain that we can maximize the reconstruction term $\mathbb{E}_{p^*(x)}\mathbb{E}_{q_\varphi(z|x)} \log r(x|G_\theta(z))$ by minimizing $-\mathbb{E}_{p^*(x)}\mathbb{E}_{q_\varphi(z|x)} \log \frac{D_\tau(x,z,G_\theta(z))}{1-D_\tau(x,z,G_\theta(z))}$ and optimize it using gradient based methods. We note again that we do not require an access to an analytic form of $r(y|G_\theta(z))$.

It is an open question what is the best choice for the $r(y|G_\theta(z))$. Our expectations from $r(y|G_\theta(z))$ are that it should encourage realistic reconstructions and highly penalize for visually distorted images. In experiments, as $r(y|x)$ we use a distribution over cyclic shifts in all directions of an image $x$. This distribution is symmetric with respect to all directions and has a mode in $x$, therefore it is the symmetric likelihood (see Definition 1 for details).

Although in practice we use $r(y|x)$ which has an explicit form due to non-optimality of $D_\tau(x,z,y)$ (that is always the case when training on finite datasets) the ratio $\log \frac{D_\tau(x,z,G_\theta(z))}{1-D_\tau(x,z,G_\theta(z))}$ sets *implicit likelihood* of reconstructions. We can think of the non-optimality of $D_\tau(x,z,y)$ as a form of regularization that allows us to convert explicit function $r(y|x)$ into implicit likelihood that has desirable properties, i.e. encourages realistic reconstructions of $x$ and penalizes unrealistic ones.

**Implicit Encoder.** The KL term from $\mathcal{L}_{\text{ELBO}}(\theta, \varphi)$ can be computed either analytically, using the Monte Carlo estimation or by the adversarial manner. We chose the latter approach proposed by Mescheder et al. (2017) because it enables implicit variational distribution $q_\varphi(z|x)$ defined by a neural sampler (encoder) $E_\varphi(x,\xi)$ where $\xi \sim \mathcal{N}(\cdot|0, I)$. For this purpose we should train a discriminator $D_\zeta(x,z)$ which tries to distinguish pairs $(x,z)$ from $p^*(x)q_\varphi(z|x)$ versus the ones from $p^*(x)p(z)$. The training objective of $D_\zeta(x,z)$ is

$$\mathbb{E}_{p^*(x)p(z)} \log D_\zeta(x,z) + \mathbb{E}_{p^*(x)q_\varphi(z|x)} \log(1 - D_\zeta(x,z)) \rightarrow \max_\zeta \tag{5}$$

$KL(q_\varphi(z|x)\|p(z))$ is a reverse KL with respect to parameters $\varphi$, therefore we can substitute it by the expression $-\mathbb{E}_{q_\varphi(z|x)} \log \frac{D_\zeta(x,z)}{1-D_\zeta(x,z)}$ (see Section 3).

**Final Objectives.** Putting it all together we arrive at the following objective:

$$\mathcal{L}_{\lambda\text{-IJAE}}(\theta, \varphi) = (1-\lambda)D_{\text{KL}}(p_\theta(x)\|p^*(x)) - \lambda\mathcal{L}_{\text{ELBO}}(\theta, \varphi) =$$

$$= -(1-\lambda)\mathbb{E}_{p_\theta(x)} \log \frac{D_{\psi^*}(x)}{1-D_{\psi^*}(x)} -$$

$$- \lambda\mathbb{E}_{p^*(x)}\mathbb{E}_{q_\varphi(z|x)} \left[\log \frac{D_{\tau^*}(x,z,G_\theta(z))}{1-D_{\tau^*}(x,z,G_\theta(z))} + \log \frac{D_{\zeta^*}(x,z)}{1-D_{\zeta^*}(x,z)}\right] \rightarrow \min_{\theta,\varphi}$$

In practice, discriminators are not optimal therefore we train our model by alternating gradients. We maximize objectives (3), (4), (5) for $D_\psi(x), D_\tau(x,z,y), D_\zeta(x,z)$ respectively and minimize $\mathcal{L}_G(\theta)$ for the generator $G_\theta(z)$, $\mathcal{L}_E(\varphi)$ for the encoder $E_\varphi(x,\xi)$ where $\mathcal{L}_G(\theta), \mathcal{L}_E(\varphi)$ are:

$$\mathcal{L}_G(\theta) = -\lambda\mathbb{E}_{p_\theta(x)} \log \frac{D_\psi(x)}{1-D_\psi(x)} - (1-\lambda)\mathbb{E}_{p^*(x)}\mathbb{E}_{q_\varphi(z|x)} \log \frac{D_\tau(x,z,G_\theta(z))}{1-D_\tau(x,z,G_\theta(z))} \rightarrow \min_\theta \tag{6}$$

$$\mathcal{L}_E(\varphi) = -\lambda\mathbb{E}_{p^*(x)}\mathbb{E}_{q_\varphi(z|x)} \left[\log \frac{D_\tau(x,z,G_\theta(z))}{1-D_\tau(x,z,G_\theta(z))} + \log \frac{D_\zeta(x,z)}{1-D_\zeta(x,z)}\right] \rightarrow \min_\varphi \tag{7}$$

## 5 EXPERIMENTS

In experiments, we evaluate generation and reconstruction ability of our model on datasets CIFAR-10 and TinyImageNet. We used a standard ResNet architecture (Gulrajani et al., 2017) for the encoder $E_\varphi(x,\xi)$, the generator $G_\theta(z)$ and for all three discriminators $D_\psi(x), D_\tau(x,z,y), D_\zeta(x,z)$. The complete architecture description for all networks and hyperparameters used in $\lambda$-IJAE can be found in Appendix, section D. To compare our method to other autoencoding methods in the best way, we also used official and publicly available code for baselines. For AGE[1] we use a pretrained model. For SVAE[2], TwoStage-VAE (2SVAE)[3] we report metrics reproduced using officially provided code and hyperparameters. For $\alpha$-GAN we also use public implementation[4] with same architecture as in $\lambda$-IJAE.

---

[1] AGE github    [2] SVAE github    [3] TwoStageVAE github    [4] $\alpha$-GAN github

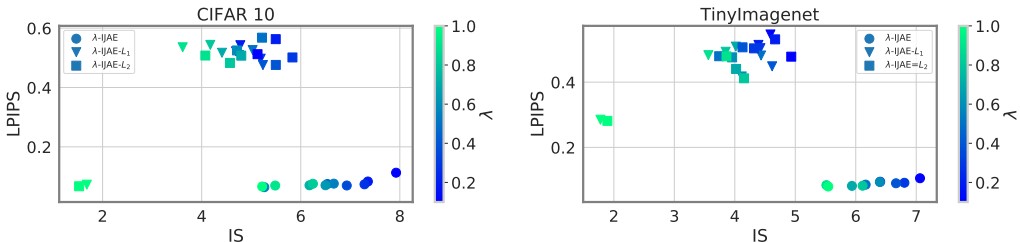

Figure 2: Comparison between $\lambda$-IJAE and $\lambda$-IJAE-$L_1$, $\lambda$-IJAE-$L_2$ models. We see that $\lambda$-IJAE results form pareto frontier with respect to IS and LPIPS for different choice of $\lambda$.

In experiments, for symmetric likelihoods $r(y|x)$ and $r'(y|x)$ we use the following: $r(y|x)$ is a continuous distribution over cyclic shifts in all directions of an image $x$. In practice, we discretize this distribution. To sample from it: (*i*) we sample one of four directions (top, bottom, right, left) equally probable; (*ii*) then sample the size of a shift (maximum size $S = 5$ pixels) from 0 to $S$ with probabilities $1/\left(\sum_{i=1}^{S} 1/(i+1)\right)$, $1/\left(2\sum_{i=1}^{S} 1/(i+1)\right)$, ..., $1/\left((S+1)\sum_{i=1}^{S} 1/(i+1)\right)$; (*iii*) as a result, we shift an image $x$ to the selected direction in a size which is sampled. For $r'(y|x)$ in practice we observe that the best choice is when $r'(y|x)$ is close to a delta function $\delta_x(y)$. Therefore, we use $r'(y|x) = \mathcal{N}(y|x, \sigma I)$ which is clearly a symmetric likelihood. We set $\sigma = 10^{-8}$. For $r(y|x)$ as an implicit likelihood we also studied a distribution over small rotations of $x$, however, we observed that cyclic shifts achieve better results.

**Evaluation.** We evaluate our model on both generation and reconstruction tasks. The quality of the former is assessed using Inception Score (IS) (Salimans et al., 2016). To calculate these metrics we used the official implementation provided in `tensorflow 1.13` (Abadi et al., 2015). The reconstruction quality is evaluated using LPIPS, proposed by (Zhang et al., 2018). LPIPS compares images based on high-level features obtained by the pre-trained network. It was show by Zhang et al. (2018) that LPIPS is a good metric which captures perceptual similarity between images. We use the official implementation (LPIPS github) to compute LPIPS.

**Ablation Study.** To show the importance of the implicit conditional likelihood $r(y|x)$ we compare $\lambda$-IJAE with its modification which has instead of implicit $r(y|x)$ a standard Gaussian or Laplace distribution. We call such models $\lambda$-IJAE-$L_2$ and $\lambda$-IJAE-$L_1$ respectively. In Figure 2 we compare $\lambda$-IJAE with $\lambda$-IJAE-$L_2$ and $\lambda$-IJAE-$L_1$ in terms of IS (generation quality) and LPIPS (reconstruction quality). We see that $\lambda$-IJAE significantly outperforms these baselines and allows to achieve pareto-optimal results for different choice of $\lambda$.

**Comparison with Baselines.** We assess generation and reconstruction quality of $\lambda$-IJAE on CIFAR-10 and TinyImageNet datasets. We compare the results to closest baselines with publicly available code. We provide visual results in Appendix, section C. Quantitative results are given in Figure 3 and in Table 1. In Figure 3 we compare the methods with respect to IS and LPIPS. Considering both metrics $\lambda$-IJAE achieves a better trade-off between reconstruction and generation quality within these

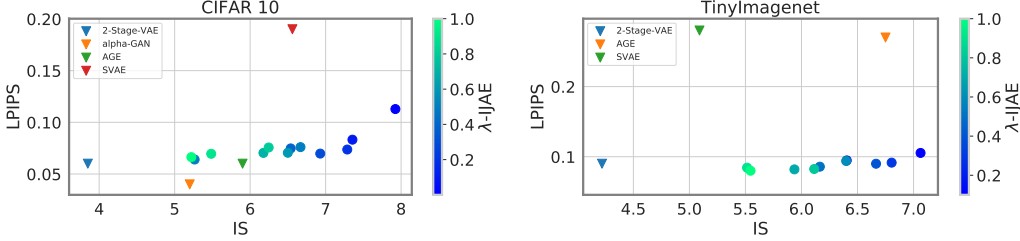

Figure 3: Evaluation of $\lambda$-IJAE on CIFAR and TinyImageNet compared to baselines. Two metrics, LPIPS and IS metrics are considered to access reconstruction and sampling quality. Considering both metrics $\lambda$-IJAE achieves a better trade-off between reconstruction and sampling quality within datasets.

Table 1: Reconstruction and generation quality on CIFAR10 and TinyImagenet for models that allow reconstructions. Baseline models were trained using publicly available code, if possible, to fill reconstruction quality metrics. $\downarrow$ – lower is better, $\uparrow$ – higher is better, best is marked with **bold**.

| Method | Generation Quality IS $\uparrow$ | Reconstruction Quality LPIPS $\downarrow$ |
|---|---|---|
| **CIFAR 10** | | |
| WAE (Tolstikhin et al., 2017) | $4.18 \pm 0.04$ | |
| ALI (Dumoulin et al., 2017)) | $5.34 \pm 0.04$ | |
| ALICE (Li et al., 2017) | $6.02 \pm 0.03$ | |
| AS-VAE (Pu et al., 2017b) | $6.3$ | |
| VAE (resnet) | $3.45 \pm 0.02$ | $0.09 \pm 0.03$ |
| 2Stage-VAE (Dai & Wipf, 2019) | $3.85 \pm 0.03$ | $0.06 \pm 0.03$ |
| $\alpha$-GAN (Rosca et al., 2017) | $5.20 \pm 0.08$ | $0.04 \pm 0.02$ |
| AGE (Ulyanov et al., 2018) | $5.90 \pm 0.04$ | $0.06 \pm 0.02$ |
| SVAE (Chen et al., 2018) | $6.56 \pm 0.07$ | $0.19 \pm 0.08$ |
| $\lambda$-IJAE ($\lambda = 0.3$) | $\mathbf{6.98 \pm 0.1}$ | $0.07 \pm 0.03$ |
| **TinyImagenet** | | |
| AGE (Ulyanov et al., 2018) | $6.75 \pm 0.09$ | $0.27 \pm 0.09$ |
| SVAE (Chen et al., 2018) | $5.09 \pm 0.05$ | $0.28 \pm 0.08$ |
| 2Stage-VAE (Dai & Wipf, 2019) | $4.22 \pm 0.05$ | $\mathbf{0.09 \pm 0.05}$ |
| $\lambda$-IJAE ($\lambda = 0.3$) | $\mathbf{6.87} \pm 0.09$ | $\mathbf{0.09 \pm 0.03}$ |

datasets. We see that small values of $\lambda$ give a good IS score while remain the decent reconstruction quality in terms of LPIPS. However, if decrease $\lambda$ further LPIPS will start to degrade. Therefore, we chose the $\lambda = 0.3$ as a reasonable trade-off between generation and reconstruction ability of $\lambda$-IJAE. For this choice of $\lambda$ we compute the results for Table 1. From these Table 1 we see that $\lambda$-IJAE achieves the state-of-the-art trade-off between generation and reconstruction quality. It confirms our justification about $\lambda$-Jeffreys divergence that it takes the best properties of both KL divergences.

## 6 CONCLUSIONS

In the paper, we considered a fusion of VAE and GAN models that takes the best of two worlds: it has sharp and coherent samples and can encode observations into low-dimensional representations. We provide a theoretical analysis of our objective and show that it is equivalent to the Jeffreys divergence. In experiments, we demonstrate that our model achieves a good balance between generation and reconstruction quality. It confirms our assumption that the Jeffreys divergence is the right choice for learning complex high-dimensional distributions in the case of the limited capacity of the model.

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

# Appendix

APPENDIX A   PROOFS

**Proposition 1.** *(Mescheder et al., 2017) Let* $D_{\psi^*}(x) = \frac{p^*(x)}{p^*(x)+p_\theta(x)}$ *for any* $x$. *Then* $-\nabla_\theta \mathbb{E}_{p_\theta(x)} \log \frac{D_{\psi^*}(x)}{1-D_{\psi^*}(x)} = \nabla_\theta D_{\mathrm{KL}}(p_\theta\|p^*)$ *and* $\nabla_\theta \mathbb{E}_{p_\theta(x)} \log(1 - D_{\psi^*}(x)) = \nabla_\theta \mathrm{JSD}(p_\theta\|p^*)$ *even if we assume that* $\nabla_\theta D_{\psi^*}(x) = 0$.

*Proof.* The core of the proof is the fact that for any distribution $q_\phi(x)$ we have $\mathbb{E}_{q_\phi(x)} \nabla_\phi \log q_\phi(x) = 0$. We can write down the derivative for the divergence as a sum of two terms: the first quantifies dependence of the divergence of parameters $\theta$ through samples $x$ and the second one quantifies dependence through log-densities. The first one can be captured by an optimal discriminator if we use pathwise gradients (aka reparametrization trick), while the second one is more challenging.

$$\nabla_\theta D_{\mathrm{KL}}(p_\theta\|p^*) = \nabla_\theta \mathbb{E}_{p_\theta(x)} \log \frac{p_\theta(x)}{p^*(x)} = \nabla_\theta \int p_\theta(x) \log \frac{p_\theta(x)}{p^*(x)} dx =$$

$$= \int \nabla_\theta p_\theta(x) \log \frac{p_\theta(x)}{p^*(x)} dx + \int p_\theta(x) \nabla_\theta \log \frac{p_\theta(x)}{p^*(x)} dx =$$

$$= \int \nabla_\theta p_\theta(x) \log \frac{1 - D_{\psi^*}(x)}{D_{\psi^*}(x)} dx + \int p_\theta(x) \nabla_\theta \log p_\theta(x) dx =$$

$$= \nabla_\theta \mathbb{E}_{p_\theta(x)} \log \frac{1 - D_{\psi^*}(x)}{D_{\psi^*}(x)} + \int p_\theta(x) \frac{\nabla_\theta p_\theta(x)}{p_\theta(x)} dx =$$

$$= \nabla_\theta \mathbb{E}_{p_\theta(x)} \log \frac{1 - D_{\psi^*}(x)}{D_{\psi^*}(x)} + \nabla_\theta \int p_\theta(x) dx = \nabla_\theta \mathbb{E}_{p_\theta(x)} \log \frac{1 - D_{\psi^*}(x)}{D_{\psi^*}(x)}$$

The same can be shown for the Jensen-Shannon divergence:

$$\nabla_\theta \mathrm{JSD}(p_\theta\|p^*) = \nabla_\theta D_{\mathrm{KL}}\left(p_\theta\|\frac{p^*+p_\theta}{2}\right) + \nabla_\theta D_{\mathrm{KL}}\left(p_\theta\|\frac{p^*+p_\theta}{2}\right) =$$

$$= \nabla_\theta \int p_\theta(x) \log \frac{p_\theta(x)}{p^*+p_\theta(x)} dx + \nabla_\theta \int p^*(x) \log \frac{p^*(x)}{p^*+p_\theta(x)} dx + \log 4 =$$

$$= \int \nabla_\theta p_\theta(x) \log \frac{p_\theta(x)}{p^*(x)+p_\theta(x)} dx + \int p_\theta(x) \nabla_\theta \log \frac{p_\theta(x)}{p^*(x)+p_\theta(x)} dx$$

$$- \int p^*(x) \nabla_\theta \log(p^*(x)+p_\theta(x)) dx + \log 4 =$$

$$= \int \nabla_\theta p_\theta(x) \log(1 - D_{\psi^*}(x)) dx - \int p_\theta(x) \nabla_\theta \log(p^*(x)+p_\theta(x)) dx$$

$$+ \int p_\theta(x) \nabla_\theta \log p_\theta(x) dx - \int p^*(x) \nabla_\theta \log(p^*(x)+p_\theta(x)) dx + \log 4 =$$

$$= \nabla_\theta \mathbb{E}_{p_\theta(x)} \log(1 - D_{\psi^*}(x)) - 2 \overbrace{\int \frac{p_\theta(x)+p^*(x)}{2} \nabla_\theta \log \frac{p_\theta(x)+p^*(x)}{2} dx}^{=0} + \log 4 =$$

$$= \nabla_\theta \mathbb{E}_{p_\theta(x)} \log(1 - D_{\psi^*}(x)) + \log 4$$

$\square$

**Theorem 1.** *Let* $D_{\tau^*}(x, z, y)$ *be the optimal solution for the objective (4) and* $r(y|x)$ *and* $r'(y|x)$ *are symmetric likelihoods. Then* $\nabla_\theta \mathbb{E}_{p^*(x)} \mathbb{E}_{q_\varphi(z|x)} \log \frac{D_{\tau^*}(x,z,G_\theta(z))}{1-D_{\tau^*}(x,z,G_\theta(z))} = \nabla_\theta \mathbb{E}_{p^*(x)} \mathbb{E}_{q_\varphi(z|x)} \log r(x|G_\theta(z)).$

*Proof.*

$$\nabla_\theta \mathbb{E}_{p^*(x)} \mathbb{E}_{q_\varphi(z|x)} \log \frac{D_{\tau^*}(x, z, G_\theta(z))}{1 - D_{\tau^*}(x, z, G_\theta(z))} = \mathbb{E}_{p^*(x)} \mathbb{E}_{q_\varphi(z|x)} \nabla_\theta \log \frac{r(G_\theta(z)|x)}{r'(G_\theta(z)|a)}|_{a=G_\theta(z)} =$$

$$= \mathbb{E}_{p^*(x)} \mathbb{E}_{q_\varphi(z|x)} \nabla_\theta \log r(G_\theta(z)|x) + \mathbb{E}_{p^*(x)} \mathbb{E}_{q_\varphi(z|x)} \left[ \nabla_\theta \log r'(G_\theta(z)|a)|_{a=G_\theta(z)} \right]$$

Now we will show the second term is equal to zero given our assumptions:

$$\mathbb{E}_{p^*(x)} \mathbb{E}_{q_\varphi(z|x)} \left[ \nabla_\theta \log r'(G_\theta(z)|a)|_{a=G_\theta(z)} \right] = \mathbb{E}_{p^*(x)} \mathbb{E}_{q_\varphi(z|x)} \frac{\frac{d}{da} r'(G_\theta(z)|a)|_{a=G_\theta(z)}}{r'(G_\theta(z)|G_\theta(z))} \nabla_\theta G_\theta(z) = 0$$

Where we have used the (1) and (2) properties of the likelihoods $r(x|y)$ (Definition 1):

$$\frac{d}{da} r'(G_\theta(z)|a)|_{a=G_\theta(z)} = \frac{d}{da} r'(a|G_\theta(z))|_{a=G_\theta(z)} = 0$$

$\square$

## APPENDIX B  FIGURE 1 SETUP

To generate the plot 1 we considered the following setup: a target distribution was a mixture:

$$p^*(x) = 0.15\mathcal{N}(x| - 8, 0.2^2) + 0.35\mathcal{N}(x| - 3, 0.8^2) + 0.3\mathcal{N}(x|3, 1) + 0.2\mathcal{N}(x|8, 0.2^2)$$

While the model as an equiprobable mixture of two learnable Gaussians:

$$p_\theta(x) = 0.5\mathcal{N}(x|\theta_1, \exp(\theta_2)) + 0.5\mathcal{N}(x|\theta_3, \exp(\theta_4))$$

The optimal $\theta$ was found by making 10,000 stochastic gradient descent iterations on Monte Carlo estimations of the corresponding divergences with a batch size of 1000. We did 50 independent runs for each method to explore different local optima and chose the best one based on a divergence estimate with 100,000 samples Monte Carlo samples.

## APPENDIX C  IMAGES

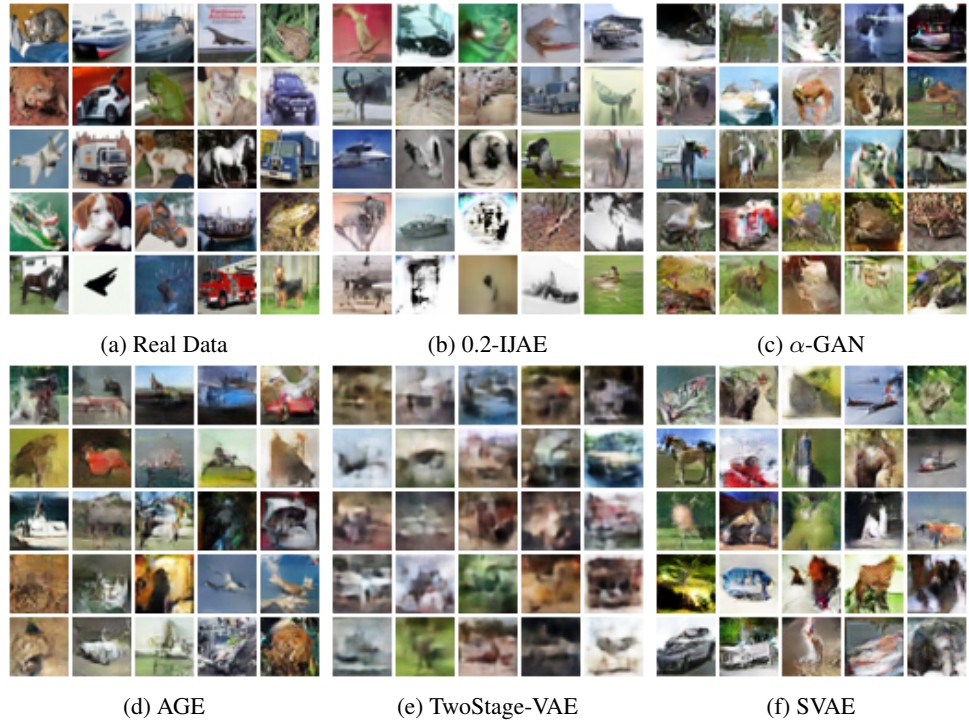

(a) Real Data      (b) 0.2-IJAE      (c) $\alpha$-GAN

(d) AGE      (e) TwoStage-VAE      (f) SVAE

Figure 4: Samples from models trained on CIFAR10 dataset. Images for baselines were obtained running publicly available code.

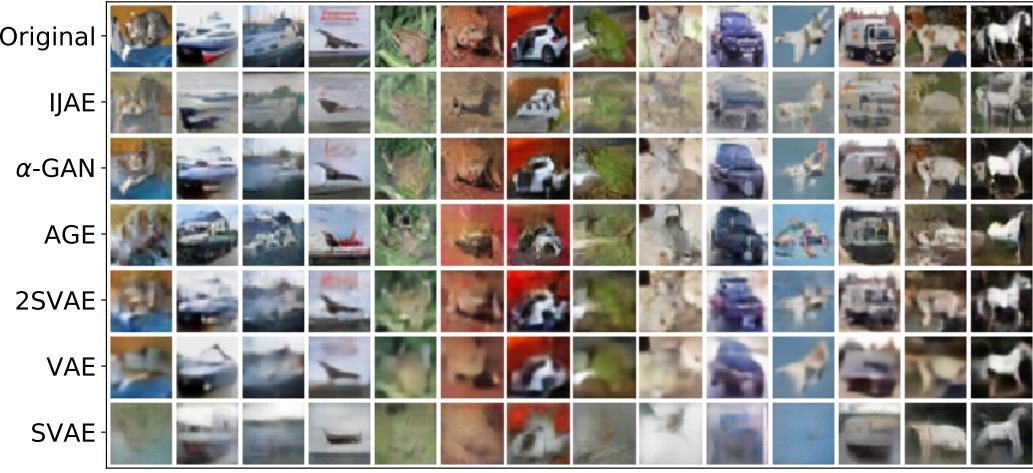

Figure 5: Reconstructions on the CIFAR10 dataset for IJAE model and closest baselines. Reconstructions for baselines were obtained running publicly available code.

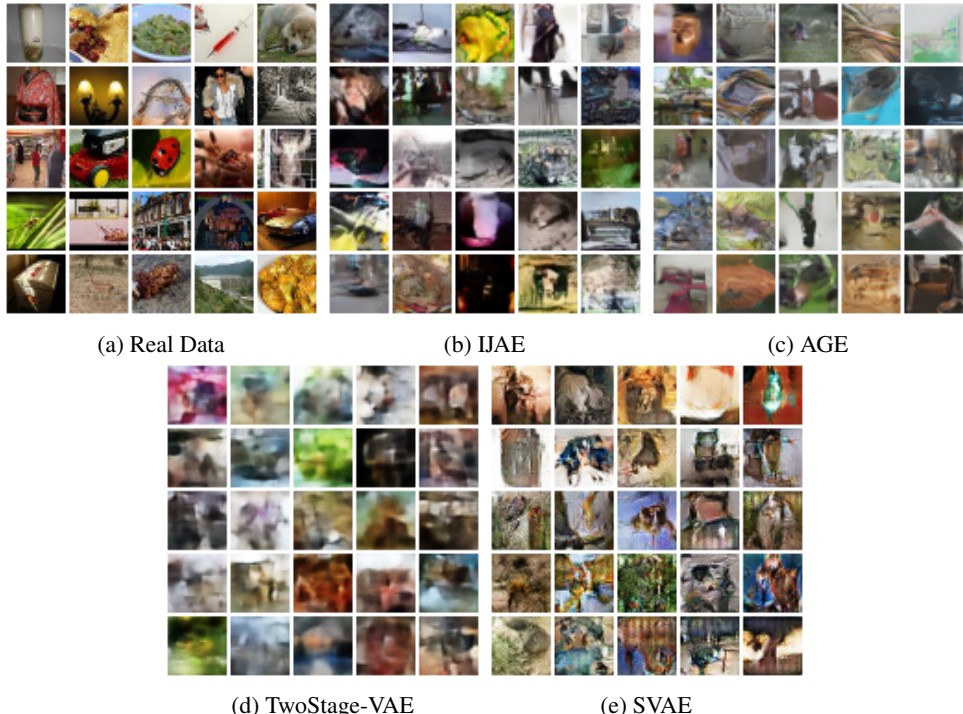

(a) Real Data        (b) IJAE        (c) AGE

(d) TwoStage-VAE        (e) SVAE

Figure 6: Samples from models trained on TinyImagenet dataset. Images for baselines were obtained running publicly available code.

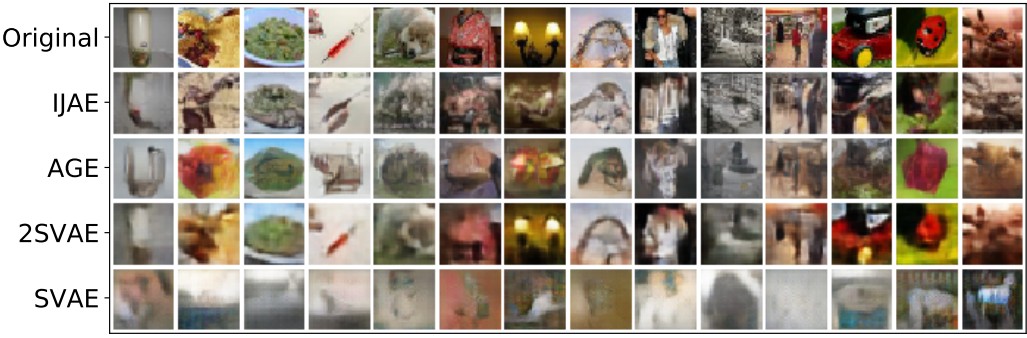

Figure 7: Reconstructions on the TinyImagenet dataset for IJAE model and closest baselines. Reconstructions for baselines were obtained running publicly available code.

## APPENDIX D    NETWORK ARCHITECTURES AND HYPERPARAMETERS

Table 2: Hyperparameters used in experiments

| Parameter | Value |
|---|---|
| Generator, Encoder Optimizer betas = (0.5, 0.9) | Adam |
| learning rate | 0.0001 |
| weight decay | 0 |
| latent dimension | 128 |
| Discriminators Optimizer betas = (0.5, 0.9) | Adam |
| learning rate | 0.0001 |
| weight decay | 0 |

Table 3: Encoder Network Architecture

| **Shortcuts** | |
|---|---|
| Basic Res Block | same as in original Resnet |
| Inject Noise | Linear(128, num channels)(normal(size=128)) + image |
| **Layer** | Parameters |
| Basic Res Block | in channels = 3, out channels = 128, stride = 2 |
| Inject Noise | |
| Basic Block | in channels = 128, out channels = 128, stride = 2 |
| Inject Noise | |
| Basic Block | in channels = 128, out channels = 128, stride = 2 |
| Basic Block | in channels = 128, out channels = 128, stride = 2 |
| ReLU | |
| Inject Noise | |
| Flatten | |
| Linear | in features 512, out features = 128 |
| **#Weights** = 6436224 | |

Table 4: Generator Network Architecture

| **Shortcuts** | |
|---|---|
| Upsample Basic Res Block | same as in original Resnet with upsample instead of strided convolution |
| **Layer** | Parameters |
| Linear | in features 128, out features = 2048 |
| Reshape to Image | shape = (128, 4, 4) |
| Upsample Basic Res Block | in channels = 128, out channels = 128, scale factor = 2 |
| Upsample Basic Res Block | in channels = 128, out channels = 128, scale factor = 2 |
| Upsample Basic Res Block | in channels = 128, out channels = 128, scale factor = 2 |
| Batch Norm | |
| ReLU | |
| Conv3x3 | stride = 1, padding 1 |
| Tanh | |
| **#Weights**=1154179 | |

Table 5: Single Image Discriminator Network Architecture

| Shortcuts | |
|---|---|
| Basic Res Block | same as in original Resnet |
| Inject Noise | Linear(128, num channels)(normal(size=128)) + image |
| **Layer** | Parameters |
| Basic Res Block | in channels = 3, out channels = 128, stride = 2 |
| Inject Noise | |
| Basic Block | in channels = 128, out channels = 128, stride = 2 |
| Inject Noise | |
| Basic Block | in channels = 128, out channels = 128, stride = 1 |
| Basic Block | in channels = 128, out channels = 128, stride = 1 |
| ReLU | |
| Global Average Pooling | |
| Linear | in features 128, out features = 1 |
| **#Weights** = 1053825 | |

Table 6: Pair Image + Latent Discriminator Network Architecture

| Shortcuts | |
|---|---|
| Basic Res Block | same as in original Resnet |
| Inject Noise | Linear(128, num channels)(normal(size=128)) + image |
| **Inputs:** | left image, right image, latent code |
| **Layer** | Parameters |
| Stack Images | orient = horizontal |
| Basic Res Block | in channels = 3, out channels = 128, stride = 2 |
| Inject Noise | |
| Basic Block | in channels = 128, out channels = 128, stride = 2 |
| Inject Noise | |
| Basic Block | in channels = 128, out channels = 128, stride = 1 |
| Basic Block | in channels = 128, out channels = 128, stride = 1 |
| ReLU | |
| Global Average Pooling | |
| Concat Latent Code | |
| Linear | in features 256, out features = 1 |
| **#Weights** = 1054081 | |

Table 7: Image + Latent Discriminator Network Architecture

| Layer | Parameters |
|---|---|
| Basic Block | kernel size = 4 in channels = 3, out channels = 64, stride = 2, padding = 1 |
| LeakyReLU(0.2) Basic Block | kernel size = 4 in channels = 64, out channels = 128, stride = 2, padding = 1 |
| LeakyReLU(0.2) Flatten | |
| Linear | in features 8320, out features = 512 |
| LeakyReLU(0.2) Linear | in features 512, out features = 256 |
| LeakyReLU(0.2) Linear | in features 512, out features = 256 |
| **#Weights** = 4526273 | |

