# OpenReview forum: "Implicit λ-Jeffreys Autoencoders: Taking the Best of Both Worlds"
_ICLR.cc/2020/Conference — Reject_

### Official Review · AnonReviewer3 · 2019-10-07
**Official Blind Review #3**

**Rating:** 3

**Review:**

The paper proposes to replace the KL-divergence in VAE training with the lambda-jeffreys divergence of which the symmetric KL-divergence is a special case. The paper proposes a pure implicit likelihood approach that uses three discriminator models to estimate the KL-divergences. Experiments are conducted on CIFAR-10 and TinyImageNet and several scores are reported to show that the proposed method performs as good if not better than current approaches.
---------
I think the paper tries to achieve too much in too little space and foregoes scientific exactness for the sake of claiming SOTA. Since there is a difference between claiming SOTA on a task and validating a new method, the small amount of space makes it difficult to substantiate both claims at the same time. In the rest of the review i will try to substantiate the claim:

1. The paper claims on page 2: "These models do not have a sound theoretical justification about what distance [...] they optimize". While the paper tries to substantiate its claims by showing theoretically that it does the right thing using the optimal discriminator, it leaves the question open what happens with any other discriminator. The theory does not justify non-optimal solutions. It is argued on page 6 that non-optimality of the discriminator serves as some form of regularization, but  this requires some justification.
Moreover, the paper uses LPIPS to measure reconstruction quality - but this measure is a deep neural network. So if those measures are good enough to compare solutions with and the theoretical justification of the proposed method is shaky in practice - why not use LPIPS for training?

2. The paper proposes the discriminator in order to allow for an implicit likelihood. However, the r-function used in the experiments does not fulfill the property of a well defined likelihood, and Theorem 1 does not hold, since technically the KL-divergence is infinity. If we ignore this by adding a small amount of Gaussian noise around the sampled cyclical shifts - like the r' used in the experiments, we can easily write down the explicit likelihood function since:

r(y|x)=\sum_i w_i N(y|Shift_i(x), \sigma)

where Shift_i is the i-th shift in the set described in the paper and w_i its probability  p(y|q). So the explicit solution of theorem 1 can be written down and another ablation study would be training the method with the explicit formulation for this KL-term(i.e. only training two discriminator models). If the results are not equivalent, this implies that the discriminator does not reach the optimum. The implications of that should be discussed regarding 1.

3.  Existing ablation studies are a bit of a straw-man: the paper compares changing r(y|x) by standard Gaussian or Laplace. However, we know that a large variance does not make any sense and almost all papers use tiny variances (e.g. in beta-VAE the beta-values tend to be very small, which is equivalent to small variances here).


---------------------------
Smaller things
- Are the experimental results all with the same architecture for encoder/generator for all results you compared to? if not, the effect of that should also be tested.

- my personal biased view on the generated images is: it looks worse than alpha-GAN. Every reconstructed image has a grey tone and the generated images also offer a strong grey palette. The details don't look better as well.

- typo inroduce->introduce


**Experience Assessment:**

I have published one or two papers in this area.

**Review Assessment: Checking Correctness Of Derivations And Theory:**

I carefully checked the derivations and theory.

**Review Assessment: Checking Correctness Of Experiments:**

I assessed the sensibility of the experiments.

**Review Assessment: Thoroughness In Paper Reading:**

I read the paper thoroughly.

---

> ### Author Response · Authors · 2019-11-14
> **Response to Review #3 (part 1/2)**
>
> Dear reviewer,
>
> We would like to thank you for the thoughtful review.
> We address your concerns below.
>
> > The theory does not justify non-optimal solutions. It is argued on page 6 that non-optimality of the discriminator serves as some form of regularization, but  this requires some justification.
>
> It is a fair question because in practice we do not have optimal discriminators and the theory does not justify this case. However, in adversarial learning it is a common practice to analyze the model under assumptions of optimal discriminators. The theoretical justification of the non-optimality case is an open question in GAN community and our paper is not aimed to solve this problem.
> We utilize the adversarial framework to learn density ratios by discriminators. When we write about the non-optimality of the discriminator as some form of regularization we follow the common assumption that non-optimal discriminator can learn smooth version of empirical density ratio. In practice, we always have finite training data and therefore the optimal discriminator will learn ratio of delta functions (because the empirical data distribution is a sum of delta functions centered in data points). So, the non-optimality of the discriminator allows us to estimate somehow the true density ratio in the case of finite data.
>
> > why not use LPIPS for training?
>
> LPIPS metric initially was proposed as a good proxy for evaluation of visual quality of reconstructions. The first reason why it is not a good idea to train LPIPS directly is that it is as Inception Score (IS) based on the outputs of the deep neural network. For the IS it was shown [1] that if we train the generator by directly maximizing IS we will end up with large IS but very unnatural generated images. We can observe the same problem for the LPIPS. The second reason is that if we train the model using LPIPS then we will not be able to use this metric for a fair comparison with other methods.
>
> > However, the r-function used in the experiments does not fulfill the property of a well defined likelihood, and Theorem 1 does not hold, since technically the KL-divergence is infinity.
>
> It is true that r-function is not a well defined likelihood. However, as we said before in the usual adversarial framework the data distribution is also not well defined density in practice. It does not prevent us from learning the smooth version of the density ratio by the non-optimal discriminator (which is always the case).
>
> > If we ignore this by adding a small amount of Gaussian noise around the sampled cyclical shifts - like the r' used in the experiments, we can easily write down the explicit likelihood function… So the explicit solution of theorem 1 can be written down and another ablation study would be training the method with the explicit formulation for this KL-term(i.e. only training two discriminator models).
>
> It is a good suggestion and we will add the comparison with this explicit likelihood in the next revision of our paper. However, this explicit r-function does not differ in principle from the standard Gaussian distribution and we are likely to obtain the same results. It is another argument why we utilize the non-optimal discriminator instead of the explicit distribution. It is similar to the standard adversarial framework that we use the discriminator to train the generator instead of the explicit distribution of the dataset which can be written down as a mixture of Gaussian distributions centered in data points and with small fixed variance.
>
> Good point. However, when we train the discriminator using r-distribution we do not expect that it will perfectly fit the density ratio of r(y|x) and r’(y|x) as in the standard GAN setting we do not expect the discriminator to exactly recover the empirical data distribution. The non-optimality of the discriminator can be thought of as a form of regularization and it allows us to learn the implicit likelihood of reconstructions defined by the non-optimal discriminator itself. We will add the comparison with the explicit likelihood you mention in the next revision of our paper to illustrate the benefits of using non-optimal discriminator.

---

> > ### Author Response · Authors · 2019-11-14
> > **Response to Review #3 (part 2/2)**
> >
> > > Existing ablation studies are a bit of a straw-man: the paper compares changing r(y|x) by standard Gaussian or Laplace. However, we know that a large variance does not make any sense and almost all papers use tiny variances (e.g. in beta-VAE the beta-values tend to be very small, which is equivalent to small variances here).
> >
> > We are sorry that we did not write many details about this ablation study, further we will add them in supplementary materials. For this experiment we considered 2 settings for learning VAE with Gaussian or Laplace conditional likelihood: 1) with constant variance; 2) with learnable variances for each pixel of the image. We observe that 2 setting with learnable variances is unstable for lambda < 1 and gives significantly worse results than 1 setting. The possible explanation is that learnable variances reweight the reconstruction loss dynamically during the training process and combined with additional weighting parameter lambda it can lead to instabilities in learning. Therefore, in Figure 2 we present results for 1 setting with constant variance. The exact value of variance used in likelihood also depends on the lambda parameter. Lambda in our experiments ranges from 0.1 to 1 with step 0.1, therefore, variance value ranges from 0.05 to 0.5 with step 0.05.
> >
> > So, we can say that our ablation study is fair and it can support the significance of the proposed implicit likelihood.
> >
> > >  Are the experimental results all with the same architecture for encoder/generator for all results you compared to? if not, the effect of that should also be tested.
> >
> > Yes, we used a standard ResNet architecture [2] as our baselines.
> >
> > [1] Shane Barratt, Rishi Sharma, A Note on the Inception Score, 2018, ICML Workshop
> > [2] Ishaan Gulrajani, Faruk Ahmed, Martin Arjovsky, Vincent Dumoulin, and Aaron Courville. Improved training of wasserstein gans. 2017, NeurIPS

---

### Official Review · AnonReviewer2 · 2019-10-23
**Official Blind Review #2**

**Rating:** 3

**Review:**

This paper introduces a model named lambda-IJAE, which combines the VAE and GAN training schemes to train a generative model achieving competitive performance. The combination of VAE and GAN is justified by its theoretical interpretation as a an optimization of the lambda-Jeffreys divergence between the real data distribution and the generation distribution. This work also introduces a reformulation of the reconstruction term of the VAE loss, allowing it to be estimated implicitly using an adversarial mechanism. Finally, the latent space of the VAE is also modelled implicitly using an adversarial mechanism, following (Mescheder et al. 2017).

I am ambivalent about this paper. The proposed implicit likelihood mechanism is very interesting, but the paper contains several weaknesses that together make me unwilling to accept it.

First of all, the paper presents itself as centered on the notion of optimizing the lambda-Jeyffreys distribution, while the main contribution is actually clearly the formulation of the implicit likelihood. The use of a weighted sum of the forward & KL divergences to train a generative model is hardly new, and has already been presented a few times (Larsen et al. 2015, Dosovitskiy & Brox 2016).

In this context the paper does not present the impact of its main contribution alone. How would behave a VAE trained solely with this implicit likelihood, but a regular Gaussian latent space and without the GAN loss? This ought to be part of the ablation study in my opinion.

Secondly, the paper discusses the issue of VAE generating unrealistic samples. This is indeed a very real issue of the VAE linked to it being trained by maximum-likelihood. However illustrating it by "blurry images" (like is done several times in the paper) is a common misconception, as while this is a very classical issue with VAEs, it is mostly unrelated to the MLE estimation.

It is rather a simple consequence of the fact that using an unweighted squared error loss to model the reconstruction of the VAE is almost always a poor model. It is equivalent to modelling the observation with a Gaussian noise of variance 1/2, which is a huge noise when considering data normalized in [0;1] or [-1;1] like is traditional to do with images. Reducing this variance to a more sensible value (like a std of 0.1 for example) or allowing the model to learn it reveals the real failure mode of the VAE generating unrealistic images, which can hardly be described as "blurry".

Similarly, the ablation study evaluates the use of L1 or L2 noise instead of the cyclic shift likelihood, but does not say what variance has been used for these, which would (as explained above) be an important parameter to take into account. If a variance of 1 was used, then the results of figure 2 are unsurprising and not insightful, as the discriminator would have merely learned to differentiate between images containing a visible Gaussian noise from images that do not.

**Experience Assessment:**

I have published one or two papers in this area.

**Review Assessment: Checking Correctness Of Derivations And Theory:**

I assessed the sensibility of the derivations and theory.

**Review Assessment: Checking Correctness Of Experiments:**

I assessed the sensibility of the experiments.

**Review Assessment: Thoroughness In Paper Reading:**

I read the paper thoroughly.

---

> ### Author Response · Authors · 2019-11-14
> **Response to Review #2**
>
> Dear reviewer,
>
> We would like to thank you for the thoughtful review. The main concern you raised is about the significance of two contributions: optimization of lambda-Jeffreys divergence and formulation of the implicit likelihood. We will address each of them below.
>
> About the optimization of lambda-Jeffreys divergence you wrote:
> > The use of a weighted sum of the forward & KL divergences to train a generative model is hardly new, and has already been presented a few times (Larsen et al. 2015, Dosovitskiy & Brox 2016).
>
> It is true that the idea of combining VAE and GAN objectives is not new. There are many approaches and we consider the closest works in related work. However, our contribution is that we do not only just propose to optimize the weighted sum of VAE and GAN losses but we provide a theoretical justification about the proposed objective and prove that under assumptions of optimal discriminators our model minimizes the lambda-Jeffreys divergence. To the best of our knowledge, there are no other papers about auto-encoder models which prove that it optimizes lambda-Jeffreys divergence. If we consider (Larsen et al. 2015, Dosovitskiy & Brox 2016) papers there were proposed to optimize the loss as a sum of VAE loss and GAN-like loss (in (Dosovitskiy & Brox 2016) there was also feature matching loss). However, in these works GAN part is not equivalent to reverse KL divergence (because they do not optimize E_{p_{theta}(x)} log[D(x)/(1 - D(x))]), therefore their losses are not equivalent to lambda-Jeffreys divergence (and authors did not analyze theoretically the corresponded divergence for their objectives).
>
> Therefore, the main significance of this contribution consists in the theoretical justification of the proposed objective.
>
> About the formulation of the implicit likelihood you wrote:
>
> > In this context the paper does not present the impact of its main contribution alone. How would behave a VAE trained solely with this implicit likelihood, but a regular Gaussian latent space and without the GAN loss? This ought to be part of the ablation study in my opinion.
>
> This experiment was a part of our ablation study. We are sorry if it was unclear from the text, we will better emphasize this experiment in the next revision of our paper.
> In experiments section you can find it in the Figure 3 and it corresponds to lambda=1 (light green circle). We see that it has good LPIPS but the worst IS compared to other values of lambda. For example, its IS is significantly worse than 0.3-IJAE which we report in Table 1. So, we can say that this implicit likelihood is beneficial when we combine it with the GAN part within lambda-Jeffreys objective.
>
> So, the significance of the implicit likelihood is that it allows us to successfully combine VAE and GAN parts in our objective in contrast to explicit likelihoods which give significantly worse results (see Figure 2). However, your second concern is about this ablation study which is illustrated in Figure 2. You wrote:
>
> > the ablation study evaluates the use of L1 or L2 noise instead of the cyclic shift likelihood, but does not say what variance has been used for these, which would (as explained above) be an important parameter to take into account. If a variance of 1 was used, then the results of figure 2 are unsurprising and not insightful, as the discriminator would have merely learned to differentiate between images containing a visible Gaussian noise from images that do not.
>
> We are sorry that we did not write many details about this experiment, further we will add them in supplementary materials. For this experiment we considered 2 settings for learning VAE with Gaussian or Laplace conditional likelihood: 1) with constant variance; 2) with learnable variances for each pixel of the image. We observe that 2nd setting with learnable variances is unstable for lambda < 1 and gives significantly worse results than 1st setting. The possible explanation is that learnable variances reweight the reconstruction loss dynamically during the training process and combined with additional weighting parameter lambda it can lead to instabilities in learning. Therefore, in Figure 2 we present results for 1 setting with constant variance. The exact value of variance used in likelihood also depends on the lambda parameter. Lambda in our experiments ranges from 0.1 to 1 with step 0.1, therefore, variance value ranges from 0.05 to 0.5 with step 0.05.
>
> So, we can say that our ablation study is fair and it can support the significance of the proposed implicit likelihood.
>
> You also claim that blurry images of VAE “is mostly unrelated to the MLE estimation”. We want to clarify that we mention blurry images is only as one example of VAE unrealistic samples. Our main claim is that MLE estimation can lead to mass-covering behaviour when the model may generate from low probability regions where samples can be very unrealistic.

---

### Official Review · AnonReviewer1 · 2019-10-28
**Official Blind Review #1**

**Rating:** 3

**Review:**

This paper proposes a new training objective for generative models that combines the objectives of VAEs and GANs. The objective is equivalent to minimizing the Jeffreys divergence (a type of f-divergence) between the true probability of the data and its probability under the model.  Furthermore, the objective comes with a knob to tradeoff the relative importance of each of the two terms.  In addition, the authors develop a implicit likelihood formulation which they claim and show empirically to outperform typical explicit formulations typically used in VAEs.

Overall, it is an interesting paper that reuses a few good ideas to develop a novel training objective. The results show that using an implicit likelihood helps (Figure 2) and that it does relatively better than either GAN or VAE approaches. I have detailed comments below about the organization of the paper, some of the experimental claims as well as a few other works which may be good to cite.


- Paper organization: I would suggest moving the related work to after the background.

- GANs and VAEs are not models per se but rather training frameworks for generative models.

- While VAEs and GANs can work on many types of data (at the very least continuous), your model seems to be developed for images. Could you make it clear what changes would be needed to apply it to non-image data?

- There are many minor grammatical errors throughout the text.

- It would be useful to provide the full algorithm somewhere (e.g., using an algorithm "box")

- Possible related work. It may be worth citing these two paper:
  - f-GAN: Training Generative Neural Samplers using Variational Divergence Minimization, NIPS'16
  - Deep Generative Learning via Variational Gradient Flow, ICML'19

- It would be useful to mention early that for IS higher is better and LPIPS lower is better.

- Even though Figures 2 and 3 (to a certain extent) seem to show that results are somewhat robust to the exact value \lambda how would you propose to set it in practice?

- Figure 3 Left (CIFAR 10), it's not absolutely clear to me that alpha-GAN and perhaps AGE isn't at least as good as your approach. The meaning of the units of the axes is a bit unclear. Do you have a particular reason to prefer your method over these in this case?

  Related: in Table 1, why are there no bolded results for CIFAR + Reconstruction?

- In Figures 5 and 7 the reconstruction of IJAE sometimes seems to be pretty far from the original image (i.e., it's not that it's blurry as for VAEs, it's that the model seems to be reconstructing a completely different image). How do you explain these results?

**Experience Assessment:**

I have read many papers in this area.

**Review Assessment: Checking Correctness Of Derivations And Theory:**

I assessed the sensibility of the derivations and theory.

**Review Assessment: Checking Correctness Of Experiments:**

I carefully checked the experiments.

**Review Assessment: Thoroughness In Paper Reading:**

I read the paper at least twice and used my best judgement in assessing the paper.

---

> ### Author Response · Authors · 2019-11-14
> **Response to Review #1**
>
> Dear reviewer,
>
> We would like to thank you for your thoughtful review and valuable suggestions. We will address each your question below:
>
> > Figure 3 Left (CIFAR 10), it's not absolutely clear to me that alpha-GAN and perhaps AGE isn't at least as good as your approach. The meaning of the units of the axes is a bit unclear. Do you have a particular reason to prefer your method over these in this case?
>
> To make it more clear: on Figure 3 x-axis and y-axis correspond to Inception Score (IS) and LPIPS metrics respectively. From this plot we can see that alpha-GAN, AGE and IJAE have comparable LPIPS. However, the advantage of our method is that we can offer a model with significantly better IS by keeping LPIPS almost the same. Particularly, for CIFAR-10 in Table 1 we chose lambda = 0.3 for which IJAE achieves the best IS and comparable LPIPS (we can see from variance bars that differences in LPIPS are insignificant). So, the main reason to prefer our method over baselines is that it can provide a much better tradeoff between generation and reconstruction qualities.
>
> > Related: in Table 1, why are there no bolded results for CIFAR + Reconstruction?
>
> The main reason is that most differences in LPIPS are insignificant due to large variance bars. In the next revision of our paper we can bold all results except ones which are significantly worse than others.
>
> > Even though Figures 2 and 3 (to a certain extent) seem to show that results are somewhat robust to the exact value \lambda how would you propose to set it in practice?
>
> From Figures 2 and 3 we see that the reconstruction quality is robust to decreasing lambda and starts to degrade only when lambda goes below 0.2. While the generation quality is very sensitive to lambda and can be improved significantly by decreasing lambda. Therefore, in practice, we recommend setting lambda around 0.3 when IJAE has the best generation ability and acceptable reconstruction quality.
>
> > In Figures 5 and 7 the reconstruction of IJAE sometimes seems to be pretty far from the original image (i.e., it's not that it's blurry as for VAEs, it's that the model seems to be reconstructing a completely different image). How do you explain these results?
>
> Such unfaithful reconstructions can be explained by the fact that we do not use an explicit pixel-wise reconstruction loss and our implicit loss can sometimes accept such reconstructions due to the underfitting of the discriminator on triples.
>
> > While VAEs and GANs can work on many types of data (at the very least continuous), your model seems to be developed for images. Could you make it clear what changes would be needed to apply it to non-image data?
>
> It is a good question. It is true that in the paper we mostly focus on images and do not mention other types of data. However, the only thing we should change for non-image data is the implementation of the implicit conditional likelihood r(y|x) which encourages the set of faithful reconstructions for the object x. For example, for images, we chose the distribution over the shifted versions of x. If we consider sequence generating model, for example, and object x is a sequence of words, we can consider r(y|x) as a distribution over sequences that are equal to x up to synonym words. Other parts of IJAE model remain the same for non-image data.
> We will add clarification about the applicability of our model to non-image data in the next revision of our paper.
>
> > Possible related work. It may be worth citing these two papers …
>
> Thank you for pointing out these papers we missed to cite. We will add them to the related work and comment on each paper.
>
> > Paper organization: I would suggest moving the related work to after the background. It would be useful to provide the full algorithm somewhere (e.g., using an algorithm "box").
> GANs and VAEs are not models per se but rather training frameworks for generative models. There are many minor grammatical errors throughout the text. It would be useful to mention early that for IS higher is better and LPIPS lower is better.
>
> Thank you for your valuable suggestions for improving paper text and organization quality. We will certainly follow them in the next revision of our paper.

---

> > ### Comment · AnonReviewer1 · 2019-11-15
> > **Thank you**
> >
> > I just wanted to acknowledge that I have read your response and I will consider it in making a final recommendation for this paper.
> >
> > Thank you!

---

### Decision · Program_Chairs · 2019-12-19

**Decision:**

Reject

**Comment:**

The paper received Weak Reject scores from all three reviewers. The AC has read the reviews and lengthy discussions and examined the paper. AC feels that there is a consensus that the paper does not quite meet the acceptance threshold and thus cannot be accepted. Hopefully the authors can use the feedback to improve their paper and resubmit to another venue.